# Poor compliance with germline testing recommendations in colorectal cancer patients undergoing molecular residual disease testing

Suzanne Schrock-Kelley[1], Vivienne Souter[1], Michael J. Hall[2], Youbao Sha[1], Urmi Sengupta[1], Adam C. ElNaggar[1], Minetta C. Liu [1] ✉ & Jeffrey N. Weitzel [1]

## Abstract

**Background** Approximately 15% of colorectal cancers (CRCs) are associated with germline mutations. There is increasing adoption of DNA-based assays for molecular residual disease (MRD) and growing evidence supporting its clinical utility, particularly for CRC by oncologists in the U.S. We assessed the uptake of germline multi-gene panel testing (MGPT) for hereditary cancer in CRC patients receiving MRD analyses in community oncology settings.

**Methods** This retrospective study included 80 patients receiving care for CRC through community oncology practices who were referred for MRD testing at a commercial laboratory (January–March 2022). Clinical data, including test requisition forms, pathology reports, and clinical notes were reviewed. Documentation of tumor microsatellite instability and/or immunohistochemical (IHC) testing for mismatch repair (MMR) deficiency, age of CRC diagnosis, family history of cancer, and any order or recommendation for MGPT were assessed.

**Results** Overall, 5/80 (6.3%) patients in the study have documented germline MGPT; 65/80 (81.3%) patients have documented MMR testing of their colorectal tumor. Among the 5 cases with abnormal MMR IHC, 2 have MGPT. Of the 33 patients meeting the 2021 National Comprehensive Cancer Network (NCCN) criteria for genetic/familial high-risk assessment, only 2 have MGPT.

**Conclusions** Our real-world data suggest that many CRC patients receiving MRD testing and meeting NCCN (v. 2021) criteria for germline MGPT may not be receiving evaluation beyond routine MMR status. Process and educational improvements are needed in community health settings to increase access and uptake of germline testing among CRC patients regardless of age at diagnosis or MMR status.

## Plain language summary

Colorectal cancer is a major health concern worldwide. Identifying patients with hereditary cancer syndromes is important to patient care as well as their family members. We reviewed health records of 80 colorectal cancer patients undergoing different laboratory testing. Only 6.3% had specific genetic testing for inherited cancer risks, even though many patients met national guidelines for this testing. This points to a gap in clinical care. Enhancing access to genetic testing in community clinics could help more people and their families understand and manage their cancer risks.

Colorectal cancer (CRC) is a leading cause of death with an incidence of approximately 1.9 million cases worldwide in 2020 and is the third most common cause of cancer-related death in the U.S[1,2]. According to the American Cancer Society, CRC-related deaths are projected to reach 52,550 in 2023[1]. Hereditary factors play an important role as individuals harboring germline variations are at an increased risk of developing CRC[3]. Studies suggest that up to 15% of CRC patients have germline mutations[3–5]. Germline multi-gene panel testing (MGPT), designed to screen for a broad array of actionable pathogenic variants (PVs), is used to inform targeted therapy for cancer patients[6–8] and can influence decisions about surgical management of CRC[9]. Lynch Syndrome (LS), the most common hereditary colorectal and endometrial cancer syndrome, is caused by clinically

[1]Natera Inc, Austin, TX, USA. [2]Department of Clinical Genetics, Cancer Prevention and Control Program, Fox Chase Cancer Center, Philadelphia, PA, USA. ✉e-mail: mliu@natera.com

actionable pathogenic germline variants in one of the four DNA mismatch repair (MMR) genes (*MLH1*, *MSH2*, *MSH6*, *PMS2*) or by a deletion in the 3′ region of *EPCAM*[10]. Screening for LS is typically achieved through DNA testing for tumor microsatellite instability (MSI) and/or immunohistochemistry (IHC) for abnormal expression of MMR proteins, followed by MGPT[11]. Polyposis syndromes also predispose to CRC[11]. Identification of individuals with inherited cancer predisposition syndromes allows for increased and targeted cancer surveillance, cancer risk reduction and prevention, and valuable information for at-risk family members in the form of cascade testing[12].

Genomic tumor testing has been widely adopted by oncologists in the U.S. to enable targeted therapies[13]. Tumor profiling can identify actionable mutations and circulating tumor DNA (ctDNA) assays may be used for molecular residual disease (MRD) assessment[14–18]. However, MRD is not designed to identify germline PVs as tumor tissue is sequenced to identify unique clonal mutations inherent to the tumor (targeting the top 16 clonal mutations), in order to design and manufacture a personalized multiplex-PCR assay to measure MRD[17]. The clinical genomic profiling assays that may be used to design the MRD assay may or may not be able to identify probable germline variants; in any event, these variants would not be included in or reported on in the MRD assay. Nonetheless, many tumor genomic profiling tests include analysis of blood samples, but primarily to correct for clonal hematopoiesis variants in cfDNA and to discern whether a variant with a high variant allele frequency is likely to be germline rather than a PV in the tumor. Notably, there is evidence pointing to the value of detecting potentially actionable germline cancer susceptibility gene PVs in the somatic (tumor) genomic data[19,20] and guidelines recommending germline validation of potentially inherited cancer risk PVs[21].

In 2009 The Evaluation of Genomic Applications in Practice and Prevention (EGAPP) Working Group recommended universal screening, stating that all newly diagnosed CRC patients should undergo testing for MSI or IHC of MMR proteins, with follow-up germline testing to be performed on MSI or MMR deficient (dMMR) tumors[22,23]. Disappointingly, translation of this guidance into clinical practice was slow, particularly in community oncology settings[24]. Nonetheless, precision oncology has driven the uptake of MSI/IHC testing as MMR status became important to predict the response to immune checkpoint inhibitors in cancer patients[6,25,26]. Still, other strategies to direct germline genetic testing are needed as MSI/IHC testing for MMR proteins will fail to identify some CRC patients with germline PVs, including those who harbor germline PVs in non-Lynch, cancer susceptibility genes (about 7% of all CRC patients)[27].

There are numerous combined tumor and germline sequencing initiatives, primarily in academic health centers[13,28]. It is unclear to what extent patients receiving care in community oncology settings have access to or decline germline testing for hereditary cancer syndromes in the context of precision oncology.

The objective of this study was to evaluate germline MGPT uptake in the context of precision oncology for CRC patients referred for ctDNA screening for MRD from community oncology settings. The need to understand potential gaps in access to MGPT for CRC patients is accentuated by the recent update of National Comprehensive Cancer Network (NCCN) guidelines to consider MGPT for all CRC patients[11].

Our results suggest that many CRC patients receiving MRD testing and meeting the updated NCCN criteria for MGPT may not be receiving evaluation beyond routine MMR status. Process and educational improvements are needed in community health settings to increase access and uptake of germline testing among CRC patients regardless of age at diagnosis or MMR status.

## Methods
### Cohort description
This retrospective cohort study included consecutive CRC patients referred from 27 non-academic community-based facilities in the Northwest U.S. for MRD testing at a commercial CLIA-approved laboratory (Natera, Inc., Austin, TX) (January–March 2022)[14–18]. A clinically validated, personalized, tumor-informed 16-plex polymerase chain reaction (mPCR)-next-generation sequencing (NGS) assay (Signatera™, Natera, Inc.) was used for the detection and quantification of ctDNA in blood samples. Briefly, a set of up to 16 patient- and tumor-specific, somatic single nucleotide variants were identified from whole exome sequencing of biopsy tumor samples and matched normal blood samples, and tracked in associated patient's blood samples. Samples with at least two tumor-specific variants detected above a pre-defined threshold were defined as ctDNA positive. ctDNA concentration was reported as mean tumor molecules per mL of plasma (MTM/mL)[17].

### Clinical data acquisition
Clinical data were abstracted from records submitted to the laboratory as part of routine test ordering. These records included the test requisition forms, pathology reports for all patients, and clinical notes where available. Biological sex was documented from the test requisition forms. Results of MMR testing (IHC/MSI) of the tumor were collected. Data on medical diagnoses, treatment history, and family history were collected, as were any notes documenting patient referral for germline MGPT (though the respective test results were not available for cases included in this study). Clinical notes were used to determine whether the patient met NCCN criteria (version 2.2021) for germline testing for hereditary cancer syndromes[11,29]. This version of the guidelines was in place when the patients in this study were enrolled.

The study was reviewed by an ethical committee (Salus IRB [#19040]) and was determined to be research-exempt due to its retrospective nature and the absence of risk to participants, with a waiver of informed patient consent.

### Reporting summary
Further information on research design is available in the Nature Portfolio Reporting Summary linked to this article.

## Results
### Study cohort characteristics
A total of 80 patients were identified and included in the study. Patients' characteristics are summarized in Table 1. Requisition forms and pathology reports were available for all patients; clinical notes were available for 96.3% (77) patients. Biological sex was fairly balanced with 55% male (44/80) and 45% (36/80) female (Table 1). The median age was 55 years (range 29–90) at the time of CRC diagnosis, and 57 years (range 29–90) at the time of MRD testing. Non-metastatic disease (stage I–III) at diagnosis was seen in 77% of the patients. At the time of MRD testing, 27% of patients were being treated for metastatic disease, 24% were in post-treatment surveillance, and 45% were receiving adjuvant therapy (Table 1).

### Analysis of collected data on MRD testing and MGPT
Overall, 5/80 (6.3%) CRC patients had documented MGPT. MMR somatic testing was documented in 65/80 (81.3%) of cases. According to the 2021 NCCN guidelines, 38 patients (47.5%) were indicated for MGPT testing; 33 due to CRC diagnosis < 50 years of age and five due to abnormal tumor IHC/MSI testing. Of the 33 early-onset patients, 18 patients were found to be MMR proficient, and the remaining (N = 15) did not have MMR testing. Of the patients who were MMR-proficient and had CRC diagnosis > 50 years, 1/42 (2.4%) had documented MGPT (Table 2). Of the five MMR-deficient patients, two (40%) had documented MGPT testing. MGPT was also obtained in one patient who did not receive MMR testing.

## Discussion
There is a potential clinical impact of germline cancer predisposition testing for treatment and risk reduction interventions for CRC patients[6–9], and the results of combined tumor and germline sequencing initiatives highlight many PV carriers that would not have been identified based on guideline-recommended testing[28,30]. Our goal was to evaluate germline genetic testing for hereditary cancer syndromes in a cohort of CRC patients receiving MRD testing as part of routine care in the community oncology setting[13,28].

Our real-world data from the community oncology setting suggest that a majority of CRC patients who received MRD testing and met the prior NCCN (v. 2021) criteria for MGPT may not have received evaluation beyond routine MMR status.

MRD testing is considered at the time of initial diagnosis of CRC for recurrence risk stratification to help predict benefit from adjuvant chemotherapy. This is also the point at which germline testing might be considered. Our study highlights that opportunities to identify germline PVs in hereditary cancer genes through MGPT in these patients may be frequently missed, particularly for clinical genomic tumor profiling assays where blood samples are not employed in paired sequencing. This gap in CRC care is particularly important since 80-85% of cancer patients in the U.S. receive their oncology care in non-academic centers[31,32]. Delayed or limited adoption of guideline-supported best practices in community settings has been noted[31,32]. A 2012 study determined that while 71% of National Cancer Institute-Designated Comprehensive Cancer Centers conducted reflex IHC/MSI for LS, only 15% of Community Hospital Cancer Programs were screening colorectal tumors for LS[24]. There is an urgent need to both highlight and bridge this gap in care, particularly since the NCCN has recently recommended that MGPT be considered in all CRC patients[33]. Expanding parallel somatic testing and germline MGPT for CRC patients could provide a solution to bridge this gap. Dissemination of guideline content and rationale to oncologists, and addressing patient concerns may

facilitate greater access to, and uptake of germline testing as part of combined somatic testing and germline MGPT for CRC patients.

All of the patients in our study were ascertained through referral for ctDNA testing for MRD, which includes somatic DNA sequencing of the tumor to identify appropriate variants to target for MRD testing. Sequencing of tumor DNA may provide information on potential germline mutations and provide an opportunity for orthogonal testing recommendations with a CLIA-certified germline assay[20]. However, tumor sequencing does not necessarily identify all germline PVs in hereditary cancer genes and the ability to identify CNVs is not established. Therefore, the providers should be cautioned about this limitation while making treatment decisions. We endorse the utility of using paired tumor and blood DNA sequencing for clinical genomic profiling in oncology[34].

In one study where both germline and somatic testing were undertaken, 30.5% of patients were documented harboring a germline PV, about a quarter of which (8.1% overall) would have been missed had somatic tumor testing been run alone[30]. MGPT also provides an opportunity to identify other germline PVs that may predispose patients for other malignancies. Among cancer patients who had somatic tumor DNA testing and were eventually found to have a germline PV, 11% were only diagnosed with the germline PVs after they had a second primary cancer[3].

In line with our findings in patients with CRC, research studies have repeatedly reported underutilization of MGPT in multiple cancer types despite national guidelines[35]. Studies in settings such as breast clinics have demonstrated that germline genetic testing is often not ordered even when cancer patients meet NCCN criteria[36–38]. Nationally as few as 6.1% of CRC patients may have undergone germline genetic testing[39]. The potential reasons that patients undergoing somatic tumor sequencing are not accessing MGPT for germline PVs are multifaceted. Providers may be focused on tumor genomic testing to enable targeted therapy of the cancer and may not understand the benefits of MGPT for the patient and also their family through cascade testing. There may be reluctance to recommend more testing for those who are already undergoing intensive medical treatment, or in patients with advanced disease and a low chance of long-term survival. Providers might also erroneously consider that the absence of tumor dMMR negates the value of MGPT, which can detect MMR PVs in some MMR proficient cases as well as PVs in non-Lynch cancer susceptibility genes (about 7% of all CRC patients)[27].

A systematic review of barriers and motivators to pursuing genomic cancer testing indicated that patients are generally very interested in genetic testing[40]. However, they found that patient barriers to cancer genetic testing included concerns about cost, confidentiality, clinical usefulness, and negative psychological impact. Patient motivators included cancer prediction, information on management, benefits to family members, and improving understanding about cancer. Anecdotally, many clinicians have complained that the complexity of guidelines has posed a barrier to clinical implementation[41]. Perhaps this barrier can be addressed to some extent by the simplification inherent in the updated NCCN and U.S. Multisociety Task Force guidelines where all CRC patients are now potential candidates for MGPT[11]. Additional barriers include limited access to genetic counseling services, particularly in rural areas, which may lead to the underutilization of MGPT in CRC. This could be addressed in part by greater availability of

## Table 1 | Characteristics of study population

| Characteristic | Study population (*N* = 80) |
|---|---|
| Median age at MRD testing (Range) | 57 years (29–90) |
| Median age at CRC diagnosis (Range) | 55 years (29–90) |
| Male *n* (%) | 44 (55) |
| Female *n* (%) | 36 (45) |
| Met NCCN criteria based upon family history alone[a] % (*n/N*) | 0 (0/25) |
| Cancer stage *n* (%) | |
| I | 8 (10.0) |
| II | 18 (22.5) |
| III | 36 (45.0) |
| IV | 17 (21.2) |
| Not known | 1 (1.2) |
| Stage of cancer journey *n* (%) | |
| Presurgery | 1 (1.2) |
| Adjuvant | 36 (45.0) |
| Metastatic | 22 (27.5) |
| Surveillance | 19 (23.7) |
| Recurrence | 1 (1.2) |
| Unknown | 1 (1.2) |

[a]Restricted to 25 cases where family history was included in the chart note.

## Table 2 | Germline testing in patients with a guideline-based indication

| Indication for germline testing for hereditary cancer testing | Documented germline testing *n* (%) |
|---|---|
| Tumor mismatch repair deficiency (*n* = 5) | 2 (40) |
| Personal history of CRC at <50 years (*n* = 33)[a] | 2 (5.6) |
| MMR-proficient and CRC diagnosis >50 years (*n* = 42) | 1 (2.5) |
| Total (CRC at any age; *n* = 80)[b] | 5 (6.3) |

*NCCN* National Comprehensive Cancer Network, *MGPT* multi-gene panel testing, *dMMR* deficient mismatch repair.
[a]In the group meeting NCCN criteria, MMR-proficient patients were 18, and patients with no MMR testing were 15.
[b]NCCN guidelines for considering CRC diagnosis < 50 years of age (NCCN guidelines,v.1.2023)[33].

digital or telephonic genetic information services for patients (including chatbots)[42] and greater use of genetic services through telehealth[43].

Strengths of our study include a representative sample in the community setting, albeit with a small sample size. Limitations include the possibility that the provider may not have documented previous germline testing or the offer of MGPT to their patients, or documentation could have been present but in a clinical record, which was not accessible to us. Nonetheless, pathology reports were available for the entire cohort, and clinical notes were available in 96% of patients.

Although the change in NCCN guidelines to consider MGPT evaluation for all patients with CRC has provoked debate[44] and amplified the gap in germline testing from that observed, the simplicity of the new guidelines may make identification of patients for testing easier[33]. While NCCN guidelines have recommended universal germline genetic testing for all patients with epithelial ovarian cancer, exocrine pancreatic cancer, and high-grade/metastatic prostate cancer, studies indicate that the reach in the oncology setting has been limited[21,33]. All patients in this study received Signatera™ (Natera, Inc.) testing[14,15,17,18]. Therefore, the ability to offer both somatic and germline testing via single requisition may make testing all CRC patients easier[14,15,17,18,45].

In conclusion, despite increasing tumor testing in the community oncology setting, our study suggests that many CRC patients may miss the opportunity to have MGPT for germline PVs. Parallel somatic and germline testing, increased access to genetic counseling through telehealth and genetic information services, greater awareness among oncologists of the importance and rationale for offering MGPT through educational efforts, and removing cost barriers for patients and their families, individually and collectively may help address this critical gap in CRC care.

## Data availability
Data will be available upon reasonable request from the corresponding author.

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

## Acknowledgements
We thank Sara Wiyrick for suggesting the study, and Melissa Maisenbacher for excellent assistance with the development of the manuscript. Funding was not applicable for this study.

## Author contributions
Conceptualization: S.S-K., V.S., J.W. Study design: S.S-K., V.S., J.W. Data curation/analysis: S.S-K. and J.W. Manuscript writing: All authors. Manuscript review and editing: S.S-K., V.S., Y.S., U.S., J.W., A.D., M.H., and M.L. Administrative, technical, or material support: S.S-K., V.S., U.S., and J.W., Supervision: S.S-K., V.S., and J.W.

## Competing interests
The authors declare the following competing interests: V.S., Y.S., U.S., A.D., M.L., and J.W. are employees of Natera, Inc. with stocks or options to own stocks.
