## [Peer Review File · Communications Medicine]

Reviewers' comments:

Reviewer #1 (Remarks to the Author):

The majority of the authors of this article titled 'Patients with Colorectal Cancer Undergoing Molecular Residual Disease Testing and Referral Rate for Germline Testing: A Precision Medicine Gap' are members of the company NATERA with extensive experience in the field of liquid biopsy, especially with interesting retrospective data, and more recently, the observational study GALAXY in colorectal cancer. Additionally, the topic they address is of particular interest in what we currently call the era of precision medicine, where the sequencing of white blood cells (WBCs) is employed to improve the results of massive sequencing and reduce the rate of false positives. The paper highlights the importance of considering not only somatic mutations but also germline mutations that can have an impact not only on the patient but also on their relatives.

Ref.: Borno HT et al. doi: 10.1001/jamaoncol.2020.0016.

Comments:

Abstract: In the Purpose section, when referring to MGPT, it is unclear whether it pertains to a panel of genes that includes both somatic and hereditary relevance, or solely a panel of genes aimed at identifying patients with alterations in germline variants. Please provide clarification on this point for better understanding.

Introduction: In the sentence "However, MRD is not designed to identify germline PVs," while this statement is true, it's worth noting that nowadays, most assays utilize paired sequencing with WBCs, and it is common practice to report these data to researchers.

Ref.: Borno HT et al. doi: 10.1001/jamaoncol.2020.0016. Gimeno-Valiente F et al. doi: 10.1016/j.esmoop.2023.102051.

The objective of the study needs clarification regarding the necessity of accessing MGPT for CRC patients, especially considering the current practice of paired plasma sequencing at diagnosis or primary tumor sequencing with WBCs, which allows identification of germline variants. It is unclear why there is a need to refer patients to MGPT when Signatera, for instance, conducts paired sequencing. Please clarify this point to better understand the study's objective.

Ref.: Henriksen TV et al. doi: 10.1158/1078-0432.CCR-21-2404.

Methods: There is a need for a better development of the methodology to gain a clearer understanding of the results. To the best of my knowledge, the Signatera assay utilizes not only tissue from the primary tumor but also WBCs for paired sequencing to obtain the 16 variants with the highest VAF to design the patient-specific assay for plasma. Therefore, I understand that this paired sequencing with WBCs provides the percentage of patients out of the 80 included in the study with mutations in the germline that could be reported and associated with hereditary syndromes.

Results: As a reviewer, I consider this article to be of interest; however, there is a need for better development of the results section as it is currently difficult to follow in its current state. Additionally, the discussion section is nearly triple the length of the results section.

When reporting that 6.3% of patients had documented MGPT, it would be beneficial to specify how many of these patients had a potentially pathogenic variant. Please clarify in the Methods section whether MGPT testing is solely for identifying mutations in the germline or also includes somatic mutations.

Discussion:

-In the sentence "Our study highlights that opportunities to identify germline PVs in hereditary cancer genes through MGPT in these patients may be frequently missed," please specify that this is particularly relevant for assays where WBCs are not employed in paired sequencing. Nowadays, the majority of assays utilize paired sequencing with WBCs to optimize results, allowing for the extraction of the percentage of patients with potentially pathogenic variants in the germline.

-In the sentence "This gap in CRC care is particularly important since 80-85% of cancer patients receive their oncology care in non-academic centers," please specify the countries or regions to which this statistic applies. In Europe, for example, the majority of patients are treated in public academic centers.

-In the sentence "However, tumor sequencing may not always detect all germline PVs in hereditary cancer genes, and the capability to identify CNVs is not yet established". It's worth noting that potentially pathogenic variants in the germline can be extracted with good coverage by consulting databases. Additionally, current paired sequencing with WBCs facilitates their identification. Ref.: Gimeno-Valiente F et al. doi: 10.1016/j.esmoop.2023.102051.

-It would be interesting to compare the recommendations regarding this matter in Europe.

Throughout my review, I have consistently found the article to be of interest. However, it is necessary to provide better specification in the Methods section to enhance comprehension, as well as to detail the results more thoroughly by indicating the patients undergoing MGPT and those in whom germline PVs were identified. Additionally, since Signatera is the assay utilized for determining MRD in these patients, I am puzzled as to why the data derived from WBC sequencing are not reported, necessitating MGPT testing. Given the company's data resources and access to germline sequencing, it would be highly intriguing if they could analyze retrospective study data in CRC patients and report findings on potentially pathogenic germline variants (in another paper, of course).

Reviewer #2 (Remarks to the Author):

Interesting descriptive report on the low prevalence of MGPT testing. It is very important to remind oncologists of the importance of performing adequate genetic counseling and genetic testing on those patients who meet the NCCN guidelines.

For this reason, the purpose of this work is important in the new era where MSI/IMS tumor testing is performed to decide on cancer therapy.

However, in my opinion, very limited data on the low performance in clinical practice has been included. It would have been nice to perform the MGPT analysis on all patients to truly know the number of hereditary syndromes that had not been diagnosed. Considering the difficulty of offering genetic counseling and testing, it would have been important to know the reason why the MGPT analysis had not been done (i.e., patient decision, not recommended by the oncologist, no genetic counseling in the center, etc).

These results are a reality in clinical practice, but it would have been compelling to obtain more information on the reasons why this happens and on what has been missed. However, it has been nicely pointed out by other studies in the discussion.

I would recommend reviewing gene names, as they must be in italics.

Point-by-point response to the reviewers

Reviewer #1 (Remarks to the Author):

The majority of the authors of this article titled 'Patients with Colorectal Cancer Undergoing Molecular Residual Disease Testing and Referral Rate for Germline Testing: A Precision Medicine Gap' are members of the company NATERA with extensive experience in the field of liquid biopsy, especially with interesting retrospective data, and more recently, the observational study GALAXY in colorectal cancer. Additionally, the topic they address is of particular interest in what we currently call the era of precision medicine, where the sequencing of white blood cells (WBCs) is employed to improve the results of massive sequencing and reduce the rate of false positives. The paper highlights the importance of considering not only somatic mutations but also germline mutations that can have an impact not only on the patient but also on their relatives.

Ref.: Borno HT et al. doi: 10.1001/jamaoncol.2020.0016.

Comments:

Abstract: In the Purpose section, when referring to MGPT, it is unclear whether it pertains to a panel of genes that includes both somatic and hereditary relevance, or solely a panel of genes aimed at identifying patients with alterations in germline variants. Please provide clarification on this point for better understanding.

Response: MGPT was categorized as 'germline' in the Introduction of the abstract section. We have now added the qualifier to MGPT in the Results and Conclusions sections of the abstract for clarity.

Introduction: In the sentence "However, MRD is not designed to identify germline PVs," while this statement is true, it's worth noting that nowadays, most assays utilize paired sequencing with WBCs, and it is common practice to report these data to researchers.

Ref.: Borno HT et al. doi: 10.1001/jamaoncol.2020.0016. Gimeno-Valiente F et al. doi: 10.1016/j.esmoop.2023.102051.

Response: Thank you for pointing out the distinction. We have added additional commentary about paired tumor and blood analyses and how they are typically used from the lens of tumor somatic profiling.

The objective of the study needs clarification regarding the necessity of accessing MGPT for CRC patients, especially considering the current practice of paired plasma sequencing at

diagnosis or primary tumor sequencing with WBCs, which allows identification of germline variants. It is unclear why there is a need to refer patients to MGPT when Signatera, for instance, conducts paired sequencing. Please clarify this point to better understand the study's objective.

Ref.: Henriksen TV et al. doi: 10.1158/1078-0432.CCR-21-2404.

Response: Please note that the statement about the limitation of MRD assays: *“However, MRD is not designed to identify germline PVs.¹⁷”* was intended to succinctly call out the distinction between tumor genomic profiling and MRD assays. In response to your helpful comments, we have added additional information to clarify this important distinction:

“However, MRD is not designed to identify germline PVs as tumor tissue is sequenced to identify unique clonal mutations inherent to the tumor (targeting the top 16 clonal mutations), in order to design and manufacture a personalized multiplex-PCR assay to measure MRD.¹⁷ The clinical genomic profiling assays that may be used to design the MRD assay may or may not be able to identify probable germline variants; in any event these variants would not be included in or reported on in the MRD assay. Nonetheless, many tumor genomic profiling tests include analysis of blood samples, but primarily to correct for clonal hematopoiesis variants in cfDNA and to discern whether a high VAF variant is likely to be germline rather than a PV in the tumor.”

There is a significant distinction between incidental observation of a probable germline PV in a tumor or paired blood sample as implemented by many commercial tumor profiling laboratories in the US versus CLIA-certified analytic sequencing assays (MGPT) designed for comprehensive detection of germline PVs. This is in part based on potential technical limitations in the tumor profiling focused approaches. For example, the read depth of sequencing may not be comparable to the germline focused assay and there are often known deficiencies in the detection of CNVs in the tumor profiling assay. The NCCN guidelines call for a germline qualified assay for the validation of presumed germline observations in tumor profiling assays. We included the following in the discussion section to emphasize the distinction:

“Sequencing of tumor DNA may provide information on potential germline mutations and provide an opportunity for orthogonal testing recommendations with a CLIA certified germline assay.²⁰ However, tumor sequencing does not necessarily identify all germline PVs in hereditary cancer genes and the ability to identify CNVs is not established.”

Methods: There is a need for a better development of the methodology to gain a clearer understanding of the results. To the best of my knowledge, the Signatera assay utilizes not only tissue from the primary tumor but also WBCs for paired sequencing to obtain the 16 variants with the highest VAF to design the patient-specific assay for plasma. Therefore, I understand that this paired sequencing with WBCs provides the percentage of patients out of the 80

included in the study with mutations in the germline that could be reported and associated with hereditary syndromes.

Response: As noted above in the response to the earlier comments, we hope it is now clear that the Signatera™ methodology is independent of the clinical genomic tumor profiling that is used to generate the MRD assay. Further, the tumor-specific clonal mutations that are used to constitute the MRD assay would not include a putative germline PV (which actually would be indicated by a high VAF of ~50%). We hope that the additional description and qualifiers about how the MRD assay is generated are adequately clarifying. Ultimately, the sample for our study was based on the clinical aspects of each case in the series with regard to the NCCN guidelines, rather than any observation of putative germline variants in the clinical genomic sequencing that was used to generate the Signatera™ assay.

Results: As a reviewer, I consider this article to be of interest; however, there is a need for better development of the results section as it is currently difficult to follow in its current state. Additionally, the discussion section is nearly triple the length of the results section.

When reporting that 6.3% of patients had documented MGPT, it would be beneficial to specify how many of these patients had a potentially pathogenic variant. Please clarify in the Methods section whether MGPT testing is solely for identifying mutations in the germline or also includes somatic mutations.

Response: We appreciate that this article was considered to be of interest. We believe that there is an important clinical practice message therein. We have made the clarification regarding the distinction between germline qualified MGPT testing and tumor profiling in the introduction. Unfortunately, the specific results of the few cases where MGPT was documented were not available.

Discussion:

-In the sentence "Our study highlights that opportunities to identify germline PVs in hereditary cancer genes through MGPT in these patients may be frequently missed," please specify that this is particularly relevant for assays where WBCs are not employed in paired sequencing. Nowadays, the majority of assays utilize paired sequencing with WBCs to optimize results, allowing for the extraction of the percentage of patients with potentially pathogenic variants in the germline.

Response: Thank you. We have added the respective qualifier, as suggested.

"Our study highlights that opportunities to identify germline PVs in hereditary cancer genes through MGPT in these patients may be frequently missed, particularly for clinical genomic tumor profiling assays where blood samples are not employed in paired sequencing."

-In the sentence "This gap in CRC care is particularly important since 80-85% of cancer patients receive their oncology care in non-academic centers," please specify the countries or regions to which this statistic applies. In Europe, for example, the majority of patients are treated in public academic centers.

Response: We have added the qualifier that this statement refers to the nonacademic centers in the US.

-In the sentence "... It's worth noting that potentially pathogenic variants in the germline can be extracted with good coverage by consulting databases. Additionally, current paired sequencing with WBCs facilitates their identification. Ref.: Gimeno-Valiente F et al. doi: 10.1016/j.esmoop.2023.102051.

Response: Please note the responses above clarifying that the paired sequencing is not central to the study design. We agree that clinical genomic tumor sequencing that is paired with analyses of blood/WBC DNA facilitates observation of potential germline PVs. We have added a sentence about the utility of using paired tumor and blood DNA sequencing for clinical genomic profiling in oncology and referenced the suggested manuscript.

"We endorse the utility of using paired tumor and blood DNA sequencing for clinical genomic profiling in oncology (Gimeno-Valiente F et al., 2023)."

-It would be interesting to compare the recommendations regarding this matter in Europe.

Response: We agree but we believe that it is beyond the scope of the current manuscript and as noted, the discussion is already rather lengthy.

Throughout my review, I have consistently found the article to be of interest. However, it is necessary to provide better specification in the Methods section to enhance comprehension, as well as to detail the results more thoroughly by indicating the patients undergoing MGPT and those in whom germline PVs were identified. Additionally, since Signatera is the assay utilized for determining MRD in these patients, I am puzzled as to why the data derived from WBC sequencing are not reported, necessitating MGPT testing. Given the company's data resources and access to germline sequencing, it would be highly intriguing if they could analyze retrospective study data in CRC patients and report findings on potentially pathogenic germline variants (in another paper, of course).

Response: Again, we hope that the responses clarifying the methodology and additional detail in the introduction, methods, and discussion provide necessary clarity for better comprehension by the reviewer. This should also explain why we did not report on WBC sequencing data since it was not available to the study.

We appreciate the reviewers' encouragement with regard to expanded studies utilizing more of the rapidly accumulating data resources at Natera. The goal of this manuscript was to highlight the relatively glaring clinical gap in guideline compliant germline testing, even though it was a relatively small cohort observation. In fact, we have already implemented measures and prompts to encourage companion germline testing concurrent with clinical genomic tumor profiling and MRD assays.

Reviewer #2 (Remarks to the Author):

Interesting descriptive report on the low prevalence of MGPT testing. It is very important to remind oncologists of the importance of performing adequate genetic counseling and genetic testing on those patients who meet the NCCN guidelines.

For this reason, the purpose of this work is important in the new era where MSI/IMS tumor testing is performed to decide on cancer therapy.

However, in my opinion, very limited data on the low performance in clinical practice has been included. It would have been nice to perform the MGPT analysis on all patients to truly know the number of hereditary syndromes that had not been diagnosed. Considering the difficulty of offering genetic counseling and testing, it would have been important to know the reason why the MGPT analysis had not been done (i.e., patient decision, not recommended by the oncologist, no genetic counseling in the center, etc).

Response: We agree that this appears to be a pervasive problem, and would ideally like to see MGPT analyses of all CRC patients, as is essentially advocated in the NCCN guidelines. We cited several tumor/germline studies that document the gap, the missing hereditary component, such that the germline information from the 80 patients in our cohort would not necessarily make the rationale for such testing more compelling.

Unfortunately, the retrospective design and anonymization did not permit survey of the critical and behavioral factors influencing any decisions regarding germline testing.

These results are a reality in clinical practice, but it would have been compelling to obtain more information on the reasons why this happens and on what has been missed. However, it has been nicely pointed out by other studies in the discussion.

Response: We agree wholeheartedly with the reviewer and appreciate that the discussion to some extent addressed the issue. As noted above, the goal of the manuscript was to highlight the clinical gap in guideline compliant germline testing and we have already implemented a

system of prompts to ordering providers to encourage guideline compliant companion germline testing.

I would recommend reviewing gene names, as they must be in italics.

Response: We have reviewed the italicization of gene names and corrected it.

REVIEWERS' COMMENTS:

Reviewer #1 (Remarks to the Author):

The authors have adequately addressed the comments. Although the results would require further development, as the authors have noted, the retrospective nature of the study and the anonymization of data prevent this. In the era of precision medicine, where high-throughput sequencing is a reality for a large percentage of cancer patients, the topic proposed in this paper is relevant. Despite its limitations, it paves the way for generating new hypotheses and papers.

Reviewer #2 (Remarks to the Author):

The authors have clarified my doubts and improved the article with the others reviewers considerations.